# SLURP: An Interactive SPARQL Query Planner

Jannik Dresselhaus[1,3], Ilya Filippov[1,3], Johannes Gengenbach[1,3],
Lars Heling[1,2][0000−0001−9668−8935], and Tobias Käfer[1,2][0000−0003−0576−7457]

[1] Institute AIFB, Karlsruhe Institute of Technology, Germany
[2] {firstname.lastname}@kit.edu
[3] {firstname.lastname}@student.kit.edu

**Abstract.** Triple Pattern Fragments (TPFs) allow for querying large RDF graphs with high availability by offering triple pattern-based access to the graphs. The limited expressivity of TPFs leads to higher client-side querying and communication costs with potentially many intermediate results that need to be transferred. Thus, the challenge of devising efficient query plans when evaluating SPARQL queries lies in minimizing these costs. Different heuristics and cost-based query planning approaches have been proposed to obtain such efficient query plans. However, we also require means to visualize, manually modify, and execute alternative query plans, to better understand the differences between existing planning approaches and their potential limitations. To this end, we propose SLURP[4], an interactive SPARQL query planner that assists RDF data consumers to visualize, modify, and compare the performance of different query execution plans over TPFs.

## 1 Introduction

**Motivation.** Since the inception of the Linked Data Fragment (LDF) framework [6] to describe Web interfaces for publishing and querying Linked Data, a variety of LDF interfaces have been proposed. These interfaces differ in their emphasis on server load, availability, and expressivity. This development also drove research in client-side query processing because less expressive LDF interfaces, such as Triple Pattern Fragment (TPF) servers, require the clients to devise efficient query plans to execute SPARQL queries. These TPF clients [1,3,5,6] rely on simple statistics to obtain efficient query plans that minimize the runtime and the number of requests during execution. To this end, the clients implement different query planning methods ranging from heuristics [1,5,6] to cost-model-based [3] approaches. However, it is difficult to understand the differences between these approaches by just comparing their execution performance on benchmark queries. Moreover, researchers might want to investigate alternative, custom query plans, which are potentially more efficient than the plans obtained by the well-known query planning approaches. For instance, for specific types of queries or RDF datasets with uncommon data distribution, the existing approaches might not find the optimal query plans, which can lead to excessive runtimes and a larger number of requests submitted to the server.

---

[4] https://people.aifb.kit.edu/zg2916/slurp/

In this demo, we present Slurp[4] to address these shortcomings. Slurp is a Web application for interactive SPARQL query planning that allows to visualize, modify, execute, and analyze the performance of execution plans for basic graph patterns over a given Triple Pattern Fragment (TPF) server. The tool is designed to help users to understand and compare different query planning approaches as well as to allow expert users to modify and optimize query plans to their needs. Moreover, Slurp can be used to support teaching students about query planning and query optimization.

**Related Work.**  In the area of relational databases, approaches to visualize query execution plans have been proposed [2] and many databases support the Explain keyword to provide information on the execution plan. However, few approaches have focused on the visualization, modification, and analysis of SPARQL query execution plans. Jakobsen et al. [4] propose the Performance Inspector and Plan Explorer (PIPE). PIPE enables the comparison of query plans devised by different federated SPARQL query engines with respect to their planning time, execution time, and the number of answers. Moreover, PIPE allows for visualizing and comparing the execution plans obtained by these engines. Similar to PIPE, Slurp enables the visualization and execution of query plans for different query planning approaches. In contrast to PIPE, Slurp does not focus on federated query processing over multiple SPARQL endpoints but on client-side query processing over single TPF servers. Furthermore, Slurp also allows users to modify execution plans and analyze their execution performance regarding runtime, requests performed, and answers produced.

## 2   System Architecture

An overview of the Slurp architecture is provided in Fig. 1. The architecture can be separated in the Web application frontend and an API provided by the backend. The source code for our demonstration is available online on GitHub[5].

**Frontend.** The frontend consist of a Main Page, an Editor page, and a Result Page. The Main Page provides an overview of the recently executed query plans. On the Editor page, users can specify a new query and the TPF server to execute the query over. As of now, queries with basic graph patterns are supported. Slurp allows the user to choose a query plan optimizer to obtain an initial query plan for the query. This enables users to (i) get an initial starting point when modifying the query plan, and (ii) compare query plans devised by different query planning approaches. For this demonstration, Slurp provides a basic *left-linear* query planner that uses the triple patterns' `count` metadata for join ordering. Moreover, Slurp also supports the query planner from nLDE[6] and the planner from CROP[7] with parameters settings used in [3]. For the initial query plan, the users can inspect the cardinalities of the individual triple patterns as well

---

[5] `https://github.com/Lars-H/slurp`

[6] `https://github.com/maribelacosta/nlde`

[7] `https://github.com/Lars-H/crop`

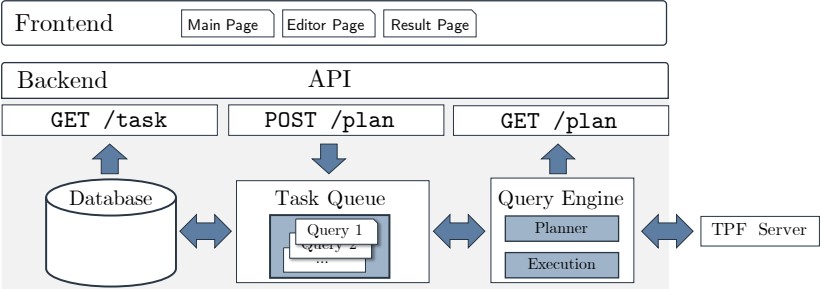

Fig. 1: Overview of the SLURP system architecture.

as the join cardinalities estimated by the query planning approach. The Editor Page provides an interactive execution plan editor that allows users to modify the initial plan or build a new plan in a drag-and-drop fashion. The leaves of the execution plan correspond to triple patterns evaluated over the selected TPF server and the inner nodes represent join operators. Users can build query plans with an arbitrary join order and place nested loop join (NLJ) or symmetric hash join (SHJ) operators in the execution plan. Upon modification, the plan can be executed and the user is redirected to the Result Page. On this page, which is automatically refreshed during query execution, the user can analyze the execution plan's performance with respect to its execution time, requests performed, intermediate result (i.e., join cardinalities), and answers produced.

**Backend.** The backend consists of three components that the frontend interacts with via an API. The query planer is used by the frontend to obtain the query plan for a given query, TPF server, and planning approach (`GET /plan`). The task queue manages the execution plans to be executed by the engine in a first-in-first-out queue (`POST /plan`). For this demonstration, a single execution plan is executed at once and we set a timeout to 60 seconds. In a local deployment of the tool, these parameters can be adjusted accordingly. The database stores the queries, execution plans, and statistics of their execution (`GET /task`).

Given a SPARQL query, a TPF server URL, and the name of an optimizer, the query planning component in the backend obtains an initial query plan which is serialized as a JSON object and provided to the frontend. The planning component currently implements the nLDE optimizer, the CROP optimizer, and a left-linear optimizer. Further optimizers may be implemented in the planning component and exposed to the frontend. When a user submits a query plan to be executed, the frontend serializes the plan as a JSON object and the SLURP backend creates the corresponding physical query plan. We use the network of Linked Data Eddies (nLDE) [1] as the engine to execute physical query plans.

**Interoperability and Reuse.** The source code of SLURP is publicly available on GitHub[5] and licensed under the open source MIT License. SLURP is developed in a containerized fashion using docker and docker-compose to facilitate installation and deployment. With the SLURP API, any application can inter-

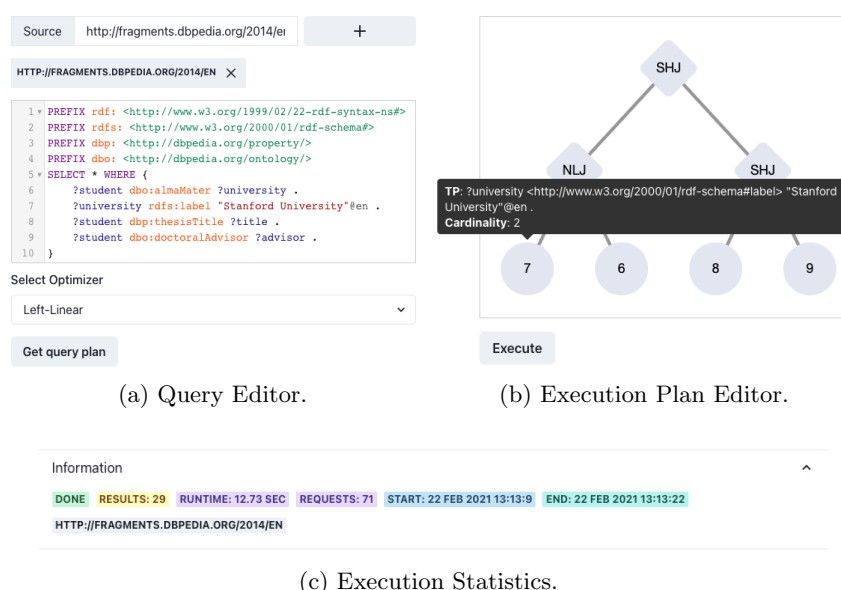

(a) Query Editor.  (b) Execution Plan Editor.

(c) Execution Statistics.

Fig. 2: SLURP: Query editor, plan visualization, and execution statistics.

act with the query planner (`GET /plan`) and the query execution engine (`POST /plan`). The query plans are represented in a binary tree with the triple patterns as the tree's leaves and join operators as its inner nodes. The query plans are serialized in JSON. In future work, we want to investigate a more generic and flexible query plan representation and serialization to support the integration of additional query planning approaches and query execution engines.

## 3  Demonstration

In the demonstration at the conference, SLURP (`https://people.aifb.kit.edu/zg2916/slurp/`) will be used to showcase different query planning approaches and how the modification of query execution plans affects their execution performance. The attendees will use SLURP to formulate SPARQL queries in the query editor over a public Triple Pattern Fragment (TPF) server (Fig. 2a). Thereafter, they will obtain an initial execution plan using one of the query plan optimizers. The attendees will be able to inspect and modify the plan in the execution plan editor (Fig. 2b): (1) changing the join order by switching triple patterns, (2) selecting nested loop or hash join operators, or (3) building a query plan from scratch by dragging and dropping sub-plans. Upon modification, the attendees will be able to execute the query plan and analyze its performance concerning its execution time, the number of performed requests, and the answers produced (Fig. 2c). In summary, the attendees will learn about client-side

query planning over TPF servers and the challenges in devising query plans that minimize the execution time and requests.

## 4    Conclusion

In this paper, we presented our demonstration of SLURP, an interactive SPARQL query planner that allows users to visualize, modify, execute, and analyze the performance of SPARQL query execution plans over Triple Pattern Fragment (TPF) servers. SLURP enables the users to compare alternative query plans obtained by different planning methods for client-side query processing. Furthermore, the users are able to modify or build query execution plans from scratch. Finally, with the query execution engine in the backend, these query plans can be executed and their performance can be compared according to the execution time, the number of requests performed, and query answers produced. In future work, we want to extend SLURP to support more features of SPARQL (e.g., filter, union, and optional expressions), additional LDF interfaces, and further query plan execution engines. Moreover, we further want to improve the tool's usability as well as facilitate the creation and comparison of alternative execution plans for the same query.

**Acknowledgement.** This work was supported by the grant QUOCA (FKZ 01IS17042) from the German Federal Ministry of Education and Research (BMBF).

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
