# OpenReview forum: "SLURP: An Interactive SPARQL Query Planner"
_eswc-conferences.org/ESWC/2021/Conference/Poster_and_Demo_Track — ESWC2021 P&D_

### Official Review · AnonReviewer4 · 2021-04-06
**An interesting paper, but for a limited audience**

**Rating:** 8
**Confidence:** 4

**Review:**

The authors want to demonstrate a tool for visualizing query plans for triple-pattern fragments. The tool aids in optimizing queries by visualizing the query plan as well as providing a means to modify and compare. An additional use case for this, not mentioned in the paper, is to educate students about query optimization. Students could play around with this and study the effects of different choices.

While I think this is a cool demo, which should be accepted for presentation, I have a few minor concerns:

1. persistence: the demo and source code are provided on systems of which it is unclear whether they will be available on the longer term.
2. I tried playing with the provided demo, but several examples timed out.
3. Modifying the execution plan is not intuitive. It became clear how it works after watching the video and after playing some more, I even found the builtin help, but it works quite unexpectedly.

A final aspect for the demo is that the target audience is a bit limited. I see an analogy with assembly code editors and visualizers.  While interesting tools, only a limited amount of people has a use for these. Most practitioners work on a higher level of abstraction.

There are also some 'cool-to-have' things that could be added:
1. The addition of the option to perform partial queries, to see the exact carnality and then plan accordingly. For the queries I tried I could also not see an estimated cardinality before executing.
2. The demo only supports BGPs. While this is a  basic building block, it does limit the scope significantly. Adding, for example, filters would make this a lot more interesting.
3. When looking at the results of a previous execution, it would be nice to be able to edit the previous plan. Currently it seems that the only way is by creating a new query.

Small bug: when there are no results to be displayed, the results table displays some JSON format. The same is downloaded when requesting the download. It seems this should be an empty table and an empty CSV file instead.

**Anonymity:**

Yes, I would like my review to remain anonymous.

---

### Official Review · ~Julien_Corman2 · 2021-04-12
**Well-designed GUI, potentially useful to test query plans for BGPs over Triple Pattern Fragments servers.**

**Rating:** 9
**Confidence:** 3

**Review:**

The article presents a graphical user interface called SLURP, that allows specifying, executing and analyzing query plans over Basic Graph Patterns (BGPs), in scenarios where the server that provides the data can only be queried via (single) triple patterns.
This setting is the one described by the "Triple Pattern Fragments (TPF) interface" specification.

In such a setting, joins are performed by issuing a sequence of triple queries to the server, where each query may be partially instantiated with results returned by previous ones.
Therefore the choice of a query plan (i.e. join typing and ordering) is key to reduce the number of queries sent to the server, and the number of triples returned by the server.

The problem is reminiscent of the one faced by SPARQL federation engines, but not immediately comparable (single VS multiple sources, triple pattern queries VS SPARQL queries, etc.).

Several planners have already been developed for this purpose.
SLURP provides a graphical interface to two of these planners (as well as a standard left-deep/nested loop planner), and the possibility to manually design and execute query plans.

Overall, the (web) GUI is of high quality for a demo, easy to use/read.
It also comes with examples, a clear documentation, and even a video tutorial.
I suspect that it will be a very welcome contribution to this line of research (in particular the visual query plan editor).
So I recommend acceptance.

As for the paper, the description of the GUI and functionalities is relatively clear.
I only wish more details were provided about the execution engine, and possibly how SLURP interfaces with existing planners (see questions and suggestions below).


## Questions

- I wonder how query planners (possibly other than the two mentioned) may interface with the GUI.
In particular, whether some serialization of query plans has been considered, or whether it may be developed in the future.

- The paper largely focuses on the GUI.
But some clarifications about the execution engine(s?) would be welcome as well.
My initial understanding was that CROP and nLDE were only used as planners, and that the authors developed their own execution engine.
But this conflicts with "executed by the engines" (Page 3, Backend), and "further query plan execution engines" (Page 4, Conclusion).

- Page 2, related work.
As far as I understand, the types of query plans that can be generated via SLURP are not SPARQL-specific.
So I wonder if similar GUIs have already been developed to design query plans graphically, for relational databases or federation engines (Dremio, Denodo, ...).

- Page 3: "challenges in devising query plans that minimize the execution time and requests while providing complete answers".
Is it possible to devise a query plan whose execution that does not return complete answers?


## Suggestions

- Intro: "higher client-side querying costs".
Communication cost as well, with a potentially large number of intermediate matches sent by the server to the client.

- Page 1: "on the servers", and Page 2: "over Triple Pattern Fragment (TPF) servers".
Maybe use "server" (singular) to avoid confusions (this is not a federation setting).

- Page 3: "with respect to its execution time, requests performed and answers produced".
I would expand this a little bit.
The interface also displays the estimated cardinalities for each join of the query plan, and the actual cardinalities after execution of the query plan, which are key information when it comes to query planning.

- Page 2 and Page 4: "analyze the performance of SPARQL query execution plans".
Maybe "BGP" instead of "SPARQL"?


## Typos

- Page 2:
"focus federated" -> "focus on federated"

- Page 2:
"federated query processing over federations of" -> "over multiple"?

**Anonymity:**

No, I would like my review to be deanonymized.

---

### Official Review · ~Ben_De_Meester1 · 2021-04-14
**Qualitative demo paper with interesting results for the TPF query plan optimization community**

**Rating:** 9
**Confidence:** 5

**Review:**

- Thank you for the very clear description of what this paper is about, web page, included video (even if there's an online demo available, this makes it very easy to review 🙂), and reproducibility of the software (MIT open source with docker containers and clear instructions).
- Clear comparison with related work. I'd be very interested to see whether there are options to combine the works (what are the hurdles to combine client-side TPF clients with server-side federated SPARQL engines?), and integrate other TPF clients such as https://comunica.dev in slurp: is there a query plan exchange format?
- In general, well-written, concise, and to-the-point demo paper which clearly describes and demonstrates the possible advantages it brings to a (maybe quite small part of) the Semantic Web community. Deserves a place at the P&D session.

**Anonymity:**

No, I would like my review to be deanonymized.

---

### Official Review · AnonReviewer2 · 2021-04-15
**interesting demo for visual query planning**

**Rating:** 8
**Confidence:** 5

**Review:**

In this demo the authors propose to manually/visually configure query plans for a Triple Pattern Fragments system (nLDE). Usually implementations use a greedy algorithm and it may be useful to change the query plan to observe specific trade offs. The negative point of manually configuring the query plan is that it only applies to a reduce set of queries, and thus what works for a query may not work for another.



**Anonymity:**

Yes, I would like my review to remain anonymous.

---

### Decision · Program_Chairs · 2021-04-19

Accept